# Obtaining Uniform High-Strength and Ductility of 2A12 Aluminum Alloy Cabin Components via Predeformation and Annular Channel Angular Extrusion

**Kai Chen** [1], **Xi Zhao** [1,2,*], **Deng-Kui Wang** [1], **La-Feng Guo** [1,*] and **Zhi-Min Zhang** [2]

1 School of Mechanical and Electrical Engineering, North University of China, Taiyuan 030051, China; 13643511918@163.com (K.C.); 15735658590@163.com (D.-K.W.)
2 National Defense Industry Innovation Center for Complex Component Extrusion Technology, Taiyuan 030051, China; nucforge@126.com
* Correspondence: zhaoxi_1111@163.com (X.Z.); 20040027@nuc.edu.cn (L.-F.G.); Tel.: +86-13934204597 (X.Z.)

**Abstract:** A 2A12 aluminum alloy component with uniform high-strength and ductility was developed via predeformation (one-pass repetitive upsetting extrusion) and annular channel angular extrusion (ACAE). Moreover, the microstructure evolution and age-hardening behavior were investigated. The results show that the upsetting-extrusion predeformation improved the cumulative strain of the component and refined the grain size, and that the second Al–Cu–Mg phases were obviously broken and refined, and that, especially, the distribution of the second phases along the extrusion direction was weakened. Thus, compared with directly ACAE-formed components, after the T6 heat treatment, the axial ultimate tensile strength (UTS) of the cabin increased from 476 to 484 MPa, and the elongation (EL) increased from 12.9% to 17.5%. The circumferential UTS increased from 426 to 482 MPa, and the EL increased from 9.24% to 16.8%. A large number of dislocations were introduced into the upsetting extrusion (UE) + ACAE method, which resulted in strain hardening and higher precipitation strengthening in the late artificial aging process. The finer and denser grains and s precipitates significantly enhanced the strength and ensured the good ductility of the alloy. It is suggested that the combination of predeformation and annular channel angular extrusion is an effective method for forming aluminum alloy cabin components with higher and more uniform mechanical properties.

**Keywords:** annular channel angular extrusion; predeformation; 2A12 aluminum alloy; heat treatment; microstructure; uniform mechanical properties

## 1. Introduction

Lightweight and high-strength aluminum alloy cabin components can meet the demands of aerospace and weapons equipment for the integration and lightweight of the major load-bearing members [1–3]. The annular channel angular extrusion (ACAE) process is a kind of large plastic deformation process for preparing cabin shells in a short process with high efficiency, which enjoys a bright prospect in the application of industrial production. Evolving from the traditional backward extrusion (BE) process, the ACAE process can form the small-diameter billet into a large-diameter cabin and shell at one time, without complicated processes such as free upsetting and punching and reaming, with a large height-to-diameter ratio. Shatermashhadi et al. [4] proposed this new method and applied it to the preparation experiment of high-purity lead components. The results show that, compared with BE, the ACAE method reduced the extrusion force in the forming process, and the prepared components had a larger equivalent strain and a more uniform distribution along the height direction, so that the final components had a uniform microstructure and performance. This new back extrusion method was expected to be applied to the preparation of ultrafine crystal samples of aluminum, copper, and

magnesium alloys. On the basis of this, S.H. Hosseini et al. [5] improved the geometric shape of the ACAE process deformation zone to obtain larger plastic strain and a more uniform distribution of strain. The improved method was applied to the preparation of industrial pure aluminum cup-shaped parts. This work focused on the microstructure evolution, the equivalent strain distribution, and the microhardness of the materials during deformation. The results show that: (1) Two shear zones provided high shear strain on the material, which led to significant grain refinement; and (2) The extrusion component had good uniformity of equivalent strain and hardness along the axial direction while obtaining a high equivalent strain and hardness. Zhao et al. [6,7] conducted a comparative study on the microstructure and properties of same-sized AZ80 magnesium alloy cabins (height: 210 mm; outer diameter: 200 mm: wall thickness: 16 mm), prepared by the ACAE method and the BE method, respectively, which showed that, compared with BE, the introduction of two strong shear deformations in the ACAE process helped refine the microstructure and significantly improved the deformation uniformity of the extruded components. This gave rise to more uniform mechanical properties along the height and thickness directions, but still exhibited anisotropy along the axial and circumferential directions. Gao et al. [8] applied the improved ACAE process for the preparation of a commercial 2A12 aluminum alloy cabin. After the T6 heat treatment, the axial UTS of the extruded cabin was increased to 476.1 Mpa, and the EL was 12.9%.

In the above works, a casting billet or extruded billet was directly used for the ACAE forming. Although the components obtained good uniformity of the strain distribution and higher mechanical properties along the axial direction, the consistency of the mechanical properties along the axial direction and the circumferential direction was ignored [9]. The cabin components not only have to withstand the load along the axial (longitudinal) direction during service but must also withstand the overload with the circumferential (transverse) direction. Highly mechanical properties with only a single direction cannot guarantee that the entire component will withstand enough outer load and keep the profile. Therefore, it is a difficult problem to prepare cabin components with the same axial and circumferential performance. The irregular second phase is usually the source of cracks and corrosion [10,11]. In this paper, the size and distribution of the second phase of an 2A12 aluminum alloy billet was controlled by upsetting-extrusion (UE) deformation. Then, the cabin was prepared by the ACAE forming, combined with a reasonable heat treatment process, and the effects of the deformation process and the heat treatment on the microstructure and mechanical properties of the 2A12 aluminum alloy were studied. The purpose was to improve the mechanical properties of the cabin and to improve the difference between the axial and circumferential performances, and this is expected to inhibit the occurrence of corrosion and the crack sources.

## 2. Materials and Methods

In this study, a commercial 2A12 aluminum alloy extrusion billet was employed, the chemical composition is shown in Table 1. A cylindrical billet with a diameter of 90 mm and a height of 360 mm was used for the experiments. The prepared billet was heated to 460 °C for 3 h, and the UE deformation of φ 90 mm-φ 140 mm-φ 90 mm was carried out to achieve the purpose of preforming the billet before the ACAE process. Then, the prepared billet was heated to 430 °C and was held for 3 h before the ACAE process.

**Table 1.** Chemical composition.

| Element | Weight % |
|---------|----------|
| Al | Others |
| Mg | 1.5 |
| Cu | 4.5 |
| Zn | $\leq 0.30$ |
| Mn | 0.4 |
| Si | 0.1 |
| Ti | $\leq 0.15$ |
| Ni | $\leq 0.10$ |

The whole process is shown in Figure 1. In the process, $MoS_2$ was used as lubricant, the extrusions were conducted at 4-THP-630, and the extrusion speed was 1 mm/s. Finally, a cabin, with a height of 160 mm, an outer diameter of 200 mm, and a wall thickness of 16 mm, was extruded. The anatomy of the cabin is shown in Figure 2a.

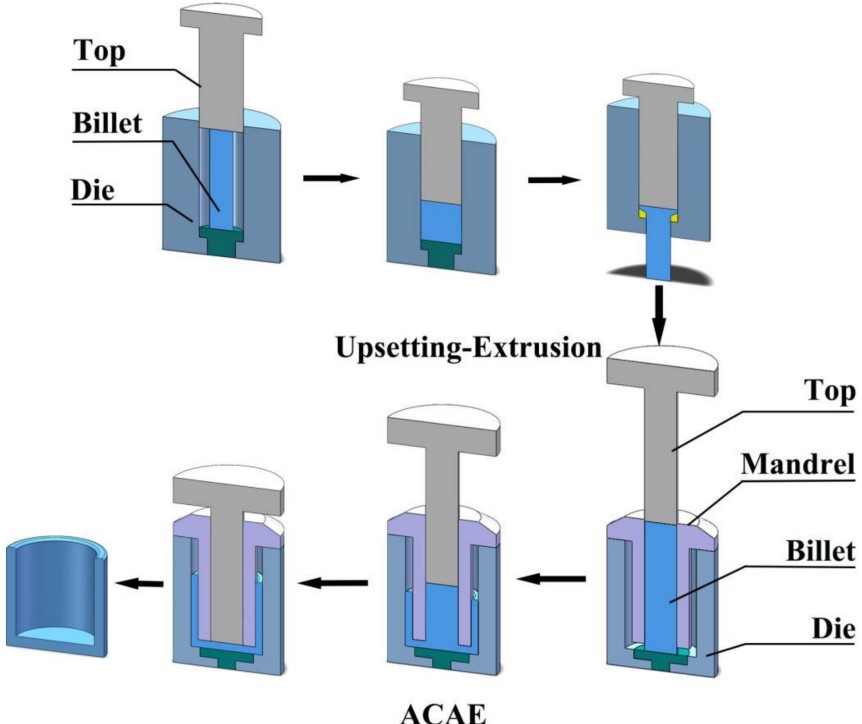

**Figure 1.** Schematic diagram of forming test.

In order to study the microstructure evolution of the material during the whole deformation process, the extrusion piece was refined into five zones, as shown in Figure 2a: (I) the billet zone; (II) Shear Zone 1; (III) the bottom zone; (IV) Shear Zone 2; and (V) the wall zone. The heat-treated samples were all taken from the wall zone, and the observation section of the samples was in an axial–circumferential direction. The sampling diagram and sample size are shown in Figure 2b. Optical microscopy (OM, Carl Zeiss, ZEIS-Image, Jena, Germany), scanning electron microscopy (SEM, Hitachi, SU-5000, Tokyo, Japan), and the electron backscattering diffraction (EBSD, AMETEK, Pegasus System, Berwyn, PA, USA) apparatus were used to observe the microstructure of the extrusion. In order to obtain more intuitive results, we compared and studied the microstructure and the properties of the extrusion billet (unupset-extruded deformation) directly applied to the ACAE-forming cabin in our team's previous research work.

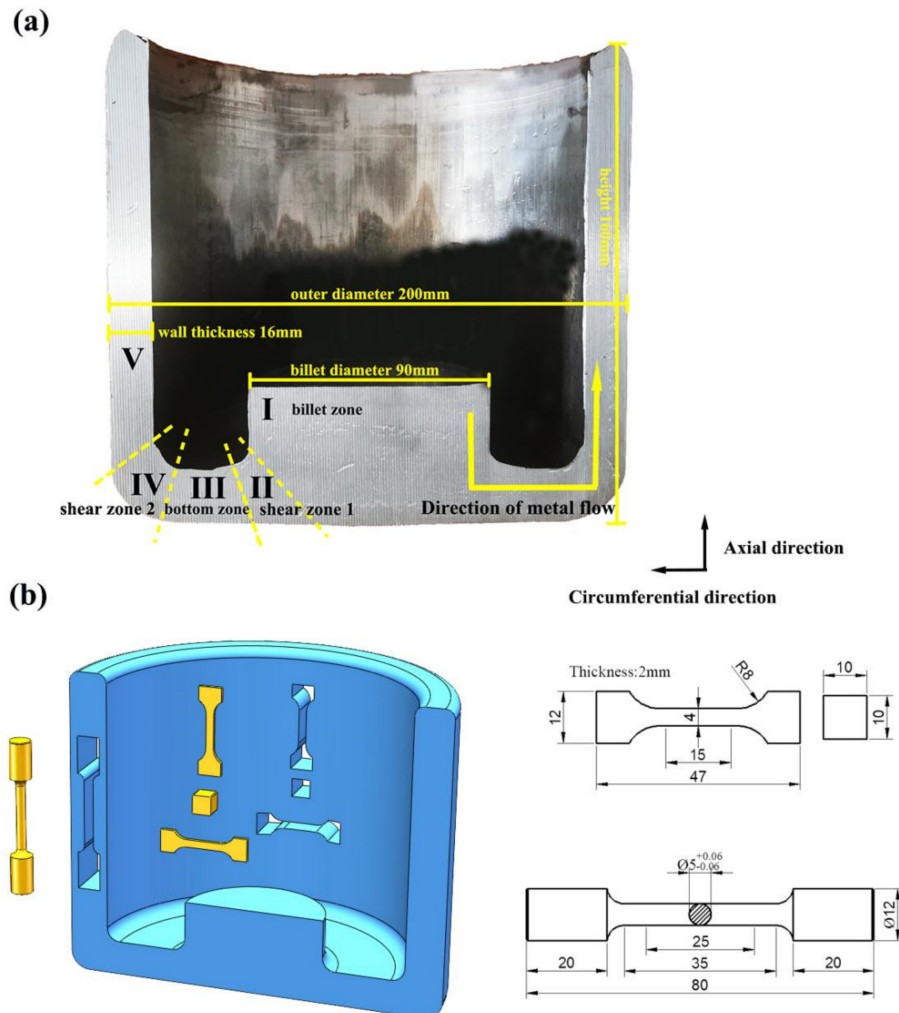

**Figure 2.** (**a**) anatomical diagram of extrusion cabin; (**b**) schematic diagram of microstructure observation and mechanical properties sampling.

## 3. Results and Discussion

### 3.1. Finite Element Simulation

In this study, Deform-3d software was used to simulate the direct ACAE- and UE + ACAE-forming processes. The mesh flow tracking method was used to represent the deformation characteristics of the metal during the forming process. In addition, the effective strain distribution of the two forming processes was shown. According to the flow grid distribution of Figure 3a, it can be seen, in Zone I, that the mesh had almost no deformation, and that the mesh shape was still the same as the square in the initial billet. After the billet passed through Shear Zone 1 (Zone II), the metal was subjected to shear force, and the mesh shape changed from a square to a parallelogram that was elongated along the diagonal. As it flowed through Zone III, the mesh continued to maintain its shape, which indicated that the metal was subjected to small strain. In Shear Zone 2 (Zone IV), the mesh was elongated along another vertical diagonal direction, which proved that the two corner zones provided shear forces that were perpendicular to each other. Finally, as shown in Figure 3b, a large strain level was accumulated in the wall area of the cabin in Zone V, and the average effective strain of the wall became 3.68.

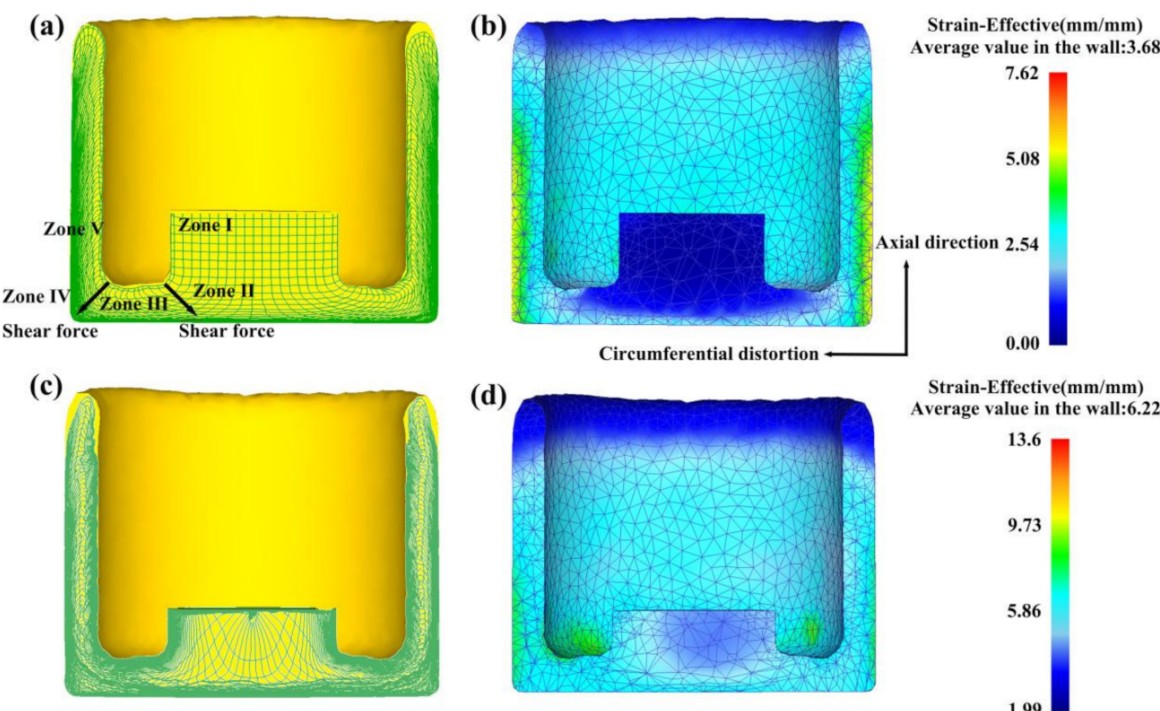

**Figure 3.** Mesh flow simulation of: (**a**) ACAE; and (**c**) UE + ACAE. Effective strain distribution simulation of: (**b**) ACAE; and (**d**) UE + ACAE.

By comparing and observing the FE simulation results of the UE + ACAE-deformed member in Figure 3c, the mesh in Zone I was found to be in an elongated shape, which was due to the advanced upsetting extrusion. In the process of deformation, the cumulative strain of the material increased, and the mesh deformation became more and more obvious. The final expression was 5.04 for the effective strain value, which was caused by the upsetting extrusion in the undeformed zone (Zone I) in Figure 3d, and it was 6.22 for the average effective strain value in the wall area (Zone V). The UE provided the initial strain and the larger cumulative equivalent strain for the subsequent ACAE process.

### 3.2. Microstructure Evolution

Figures 4 and A1 shows the BSE image of the material during the whole deformation process. The energy dispersive spectrometer (EDS) shows that the bright white parts are mainly the Al–Cu–Mg phase (Figure 5) [12,13]. Figure 4a shows, in the coarse second phase, which is often agglomerated, that the microstructure of the 2A12 extrusion bar billet is unevenly distributed, with sizes in these phases that were mostly 15–20 μm.

However, large, irregular, and inhomogeneous component phases (especially insoluble component phases containing impurities such as Fe and Si) often become the source of crack initiation and corrosion, and may promote crack propagation, thereby affecting the performance of the components [12]. In addition, a large number of rod-shaped T phases ($Al_{20}Cu_2Mn_3$) [14] with sizes below 1 μm can be clearly observed from the higher multiples BSE diagram. The T-phase particles in the 2A12 aluminum alloy precipitate in the process of the homogenization heat treatment, and gradually grow up with the extension of the homogenization time. These particles are very stable in the subsequent solid solution and aging heat treatment process, and their main effect is to prevent grain boundary slippage, and to play the role of high-temperature strengthening during the high-temperature heat treatment or deformation heat treatment. The large-sized second-phase particles usually belong to the brittle phase, and its deformation ability for the coordination of the Al matrix is poor. When the particles have irregular shapes, large sizes, and an inhomogeneous distribution, especially, they are often broken during the deformation.

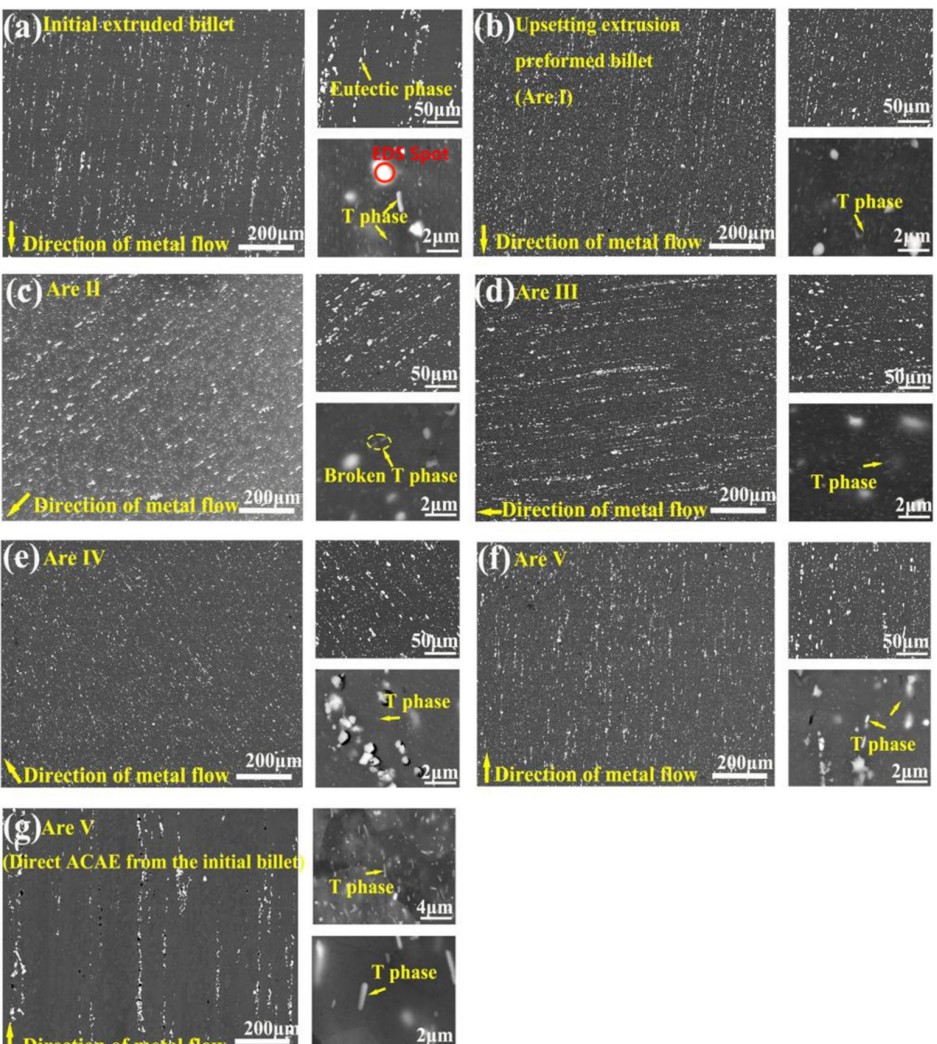

**Figure 4.** BSE image of the second-phase evolution during deformation: (**a**) initial billet; (**b**) billet deformed by UE; (**c**) UE + ACAE Zone II; (**d**) UE + ACAE Zone III; (**e**) UE + ACAE Zone IV; (**f**) UE + ACAE Zone V; and (**g**) ACAE Zone V.

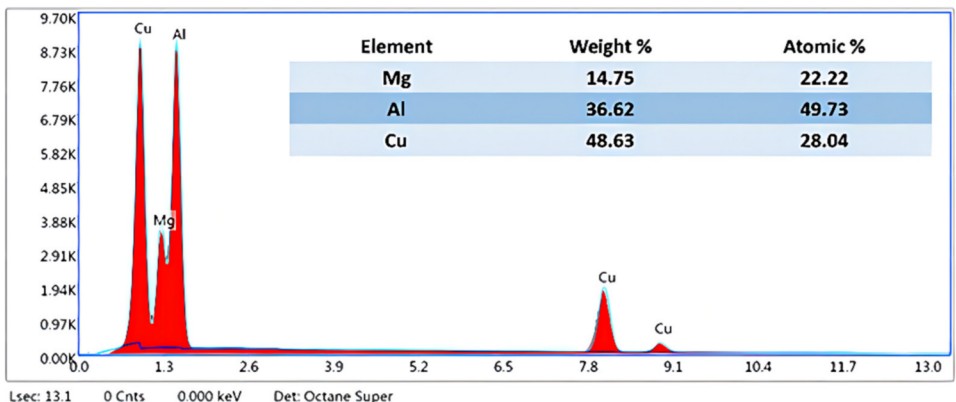

**Figure 5.** EDS results for the massive phases.

As is shown in Figure 4b, the size and distribution of the second phase in the billet were regulated by the UE. During the hot extrusion process, the component phases were broken down to less than 10 μm by the UE and were then dispersed again. Compared with

the initial extrusion bar billet, the distribution of the component phases along the extrusion direction was weakened to a large degree, and the distribution of the rod-shaped T phase was more dispersed. After the predeformation, the size of the second phase in the blank was refined, the shape tended to be regular, and the distribution was dispersive, and it provided a large cumulative strain on the subsequent ACAE deformation.

Figure 4c,e show two shear zones, Zone II and Zone III, in the process of ACAE forming. The metal was subjected to two strong shear stresses in these two zones. Although a small amount of aggregation in the second phase occurred when it passed through Zone III (Figure 4d), the direction of the strong shear stress provided by Zone IV was mutually perpendicular to that provided by Zone II, and so the degree of dispersion of the second phase increased greatly in the deformation process. In this process, the component phases were broken and refined to a diameter of less than 10 μm, with a high degree of dispersion. The broken T phases, most of which are 0.2–0.3 μm in size, were observed in Zone II. As shown in Figure 4f, the microstructure with a small size and a high degree of dispersion of the second phase was finally obtained in the wall area of Zone V. The second phase of the crushing dispersion increased the deformation capacity of the coordinated Al matrix in all directions.

Figure 4g shows the wall area of the cabin that was obtained by the direct ACAE process of the same initial billet. Compared with Figure 4f, the second phase showed obvious intergranular agglomeration distribution, and the T-phase particles with the size of 2 μm were not fully broken.

The UE can effectively break the second phase and weaken the second-phase agglomeration along the extrusion direction. The fine dispersive distribution of the second phase can be continued in the subsequent Severe Plastic Deformation (SPD) process. Compared with direct ACAE-forming components, UE + ACAE-forming components represent a higher degree of second-phase refinement and dispersion.

In order to further analyze the microstructure changes of the sample after the UE deformation and ACAE forming, an EBSD analysis was used in the microstructure of the billet and in the Zone V microstructure of the cabin, as shown in Figures 6–8. Figure 6a shows the initial microstructure of the extruded billet; Figure 6c shows the Zone V structure of the direct ACAE-forming cabin; Figure 6b shows the microstructure of the billet after the UE process; and Figure 6d shows the Zone V structure of the UE + ACAE-forming cabin. The low-angle grain boundary (LAGB,2°–15°) and the high-angle grain boundary (HAGB,15°–180°) were identified by misorientation angle quantification and are represented by the white line and black line, respectively. Figures 7 and 8 show the corresponding grain size distribution and dynamic recrystallization grain distribution.

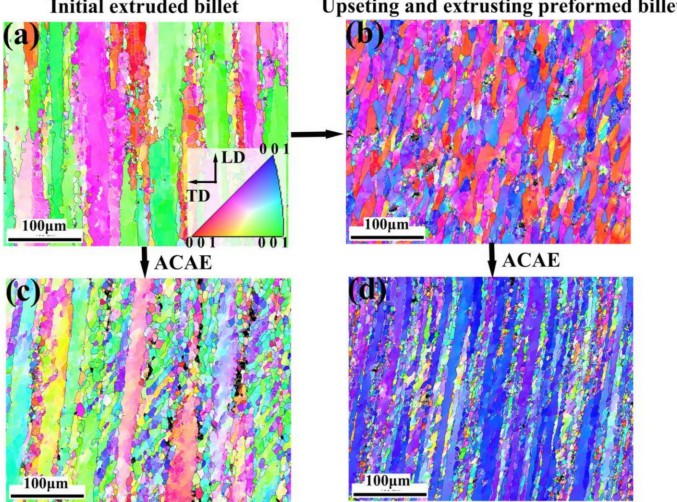

**Figure 6.** The inverse pole fig (IPF) of: (**a**) initial billet; (**b**) billet deformed by UE; (**c**) Zone V of direct ACAE-forming cabin [7]; (**d**) Zone V of UE + ACAE-forming cabin.

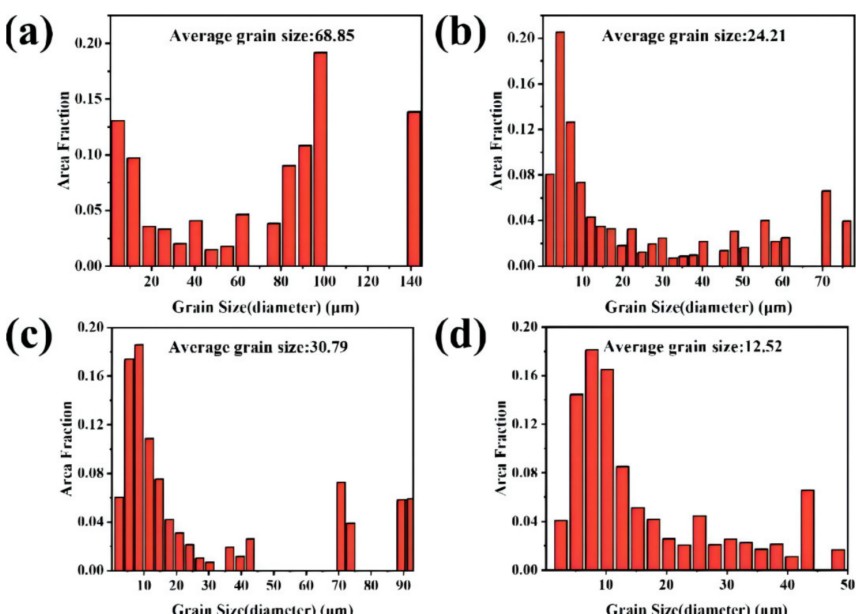

**Figure 7.** Grain size distribution of: (**a**) initial billet; (**b**) billet deformed by UE; (**c**) Zone V of direct ACAE-forming cabin; (**d**) Zone V of UE + ACAE-forming cabin.

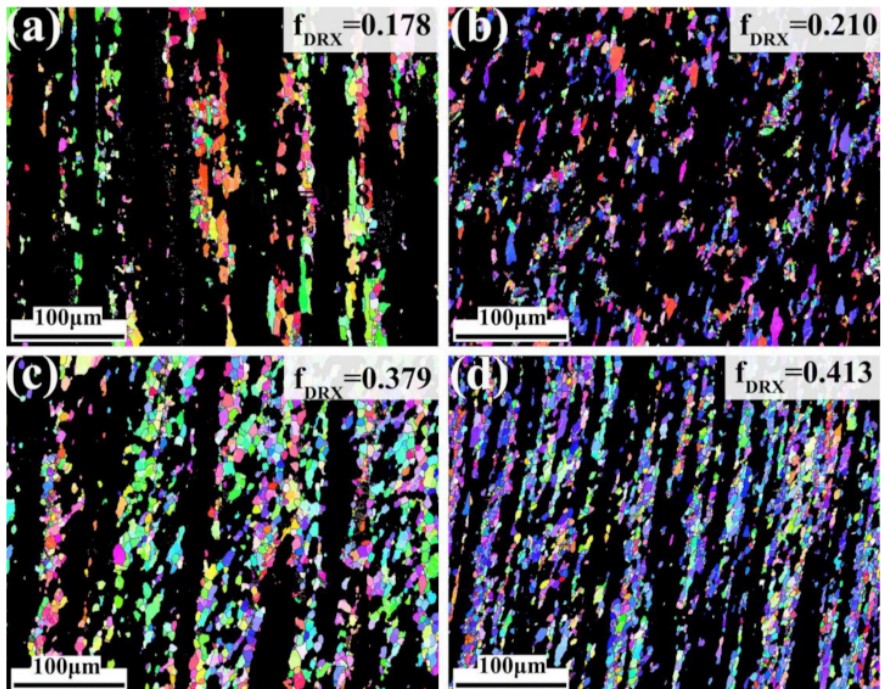

**Figure 8.** Dynamic recrystallization grain distribution of: (**a**) initial billet; (**b**) billet deformed by UE; (**c**) Zone V of direct ACAE-forming cabin; (**d**) Zone V of UE + ACAE-forming cabin.

As can be seen from Figure 6, compared with the direct ACAE structure, the internal colors of the large grains in the component structure formed by the UE + ACAE changed dramatically. Large grains turned into more elongated strips, which were surrounded by a large number of small dynamically recrystallized (DRXed) grains. This is because the UE process provided a larger cumulative strain for the whole deformation process (the average effective strain increased from 3.68 to 6.22), which made the elongation of the grains and the rotation of the lattice more intense. When the force was constantly applied

in the process of thermal deformation, plastic deformation occurred in the metal with the applied force, which resulted in the intracrystalline dislocation accumulation [15].

Along with the continuous strain accumulation and the thermal deformation, the dislocation density increased, which resulted in the recovery process. When the dislocation proliferation rate and disappearance rate reached equilibrium, a stable state was achieved. At this time, the dislocation was mainly concentrated on the cell wall to form subgrains, and then constantly evolved into new dynamic recrystallized grains, which was a typical continuous dynamic recrystallization (cDRX) process [16,17]. In the end, the two deformation structures of the ACAE cabin and the UE + ACAE cabin were both banded structures that were composed of elongated deformed grains and relatively fine DRXed grains.

According to Figure 7, the average grain size of the blank was refined from 68.85 to 30.79 μm by the direct ACAE forming, but a small number of large ones with sizes of 90 μm still existed. After the UE, the average grain size of the billet was refined to 24.21 μm, and the maximum was reduced by half compared with the initial billet. The average grain size was further refined to 12.52 μm, and its distribution was more uniform after the UE + ACAE. This shows that the large grains can be broken significantly by upsetting extrusion, and that the grain size can be further reduced in the subsequent ACAE forming process.

The grains with a Grain Orientation Spread (GOS) less than 2° were defined as DRXed grains and were displayed separately to obtain the dynamic recrystallized grain distribution, as shown in Figure 8. Meanwhile, the corresponding DRX ratio was calculated. As the initial billet was extruded, the dynamic recrystallization ratio was 17.8%. The DRX ratio (37.9%) was obtained by the direct ACAE forming. In contrast, the DRX ratio of the UE + ACAE Zone V structure was as high as 41.3%, and the DRXed grain size was significantly smaller. This may be because the billet, after the upsetting extrusion, had a finer grain size and more grain boundaries, which provided a favorable region for recrystallization nucleation in the subsequent ACAE deformation process. Moreover, at the initial stage of thermal deformation, the second-phase particles could accelerate the DRX process through particle-stimulated nucleation (PSN) [18,19]. For the UE + ACAE-forming component, more dislocation pinning points and nucleation sites could be generated by thinning and dispersing the second-phase particles. At the same time, the large cumulative strain could store more deformation energy in the material, and it generated more dynamic recrystallized grains through grain boundary migration or subgrain rotation [15–17]. Moreover, due to the Zener resistance effect, the broken fine particles impeded the grain boundary movement and inhibited the growth of DRXed grains [18,20]. Therefore, the fine microstructure after large plastic UE deformation enabled the subsequent ACAE-forming members to obtain fine DRXed grains [15].

### 3.3. Heat Treatment Microstructure

In order to improve the mechanical properties of the cabin, the UE + ACAE-formed component in this paper was conducted with a T6 heat treatment. The specific parameter was as follows: 515 °C × 1 h solid solution + 190 °C × 6 h artificial aging. The microstructure of the heat treatment is shown in Figure 9. After the T6 heat treatment, part of the residual insoluble phases was still arranged along the crystal, most of the eutectic phases were dissolved back, and the stable structure of the spiculate S phases (Al2CuMg) precipitated during the aging treatment of the 2A12 aluminum alloy [21–25]. A high-temperature solution treatment led the alloy to static recovery, the subgrains merged into recrystallized grains, and the number of HAGBs increased, which resulted in the grain boundary strengthening of the alloy. After the high-temperature solution treatment, the average grain size increased slightly to 18.46 μm; however, at the same time, the equiaxed degree of the recrystallized grains increased. This uniform structure can improve the deformation uniformity of the component in different directions after heat treatment. Moreover, several UE + ACAE plastic deformations resulted in the fine dispersion of the Al–Cu–Mg-soluble components, and a large number of subgrain boundaries and high-density dislocations

provided diffusion channels for the migration of the Cu atoms, which caused the Cu-rich phases to be more fully redissolved during the solution process; at the same time, a large number of dispersed nucleation sites were provided for the S phases, and a large number of dislocations and vacancies served as diffusion channels of the Cu atoms to promote the precipitation of the aging strengthening phase [26,27]. Finally, the difference between the axial and circumferential mechanical properties of the components was reduced after the heat treatment.

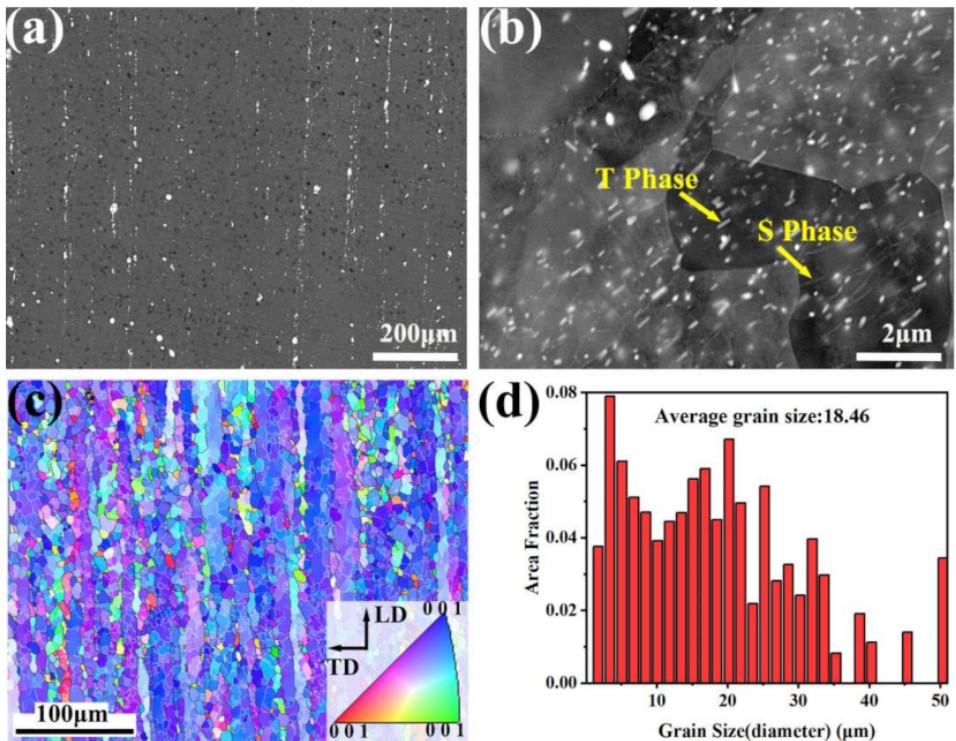

**Figure 9.** Heat treatment microstructure of UE + ACAE-forming cabin Zone V: (**a**) BSE Figure 1; (**b**) BSE Figure 2; (**c**) IPF figure; (**d**) grain size distribution.

*3.4. Mechanical Properties*

Figure 10 shows the average mechanical properties of the directly ACAE-forming cabin and the UE + ACAE-forming cabin. Axial-direction and circumferential-direction samples were used to test the mechanical properties of Cabin Zone V, according to the extrusion state and the T6 state, respectively, and the results are shown in Figure 10.

After, the billet was formed by the direct ACAE process, and the extruded mechanical properties are shown in Figure 10a. The extruded axial UTS reaches 340 MPa, the YS is 223.7 MPa, and the EL is 16.2%, while the circumferential properties are relatively low, with the UTS at 266 MPa, the YS at 155 MPa, and the EL at 9.3%. As can be seen from Figure 10b, the performance of the cabin after the T6 heat treatment shows that it maintains this characteristic. The UTS is 476 MPa, the YS increased to 371 MPa, and the EL is 12.9%. Meanwhile, the circumferential UTS is lower than the longitudinal UTS (426 MPa), the YS is 317 MPa, and the EL is 9.24%. In general, the mechanical properties of the component that was obtained by direct ACAE forming are significantly different in the axial and circumferential directions.

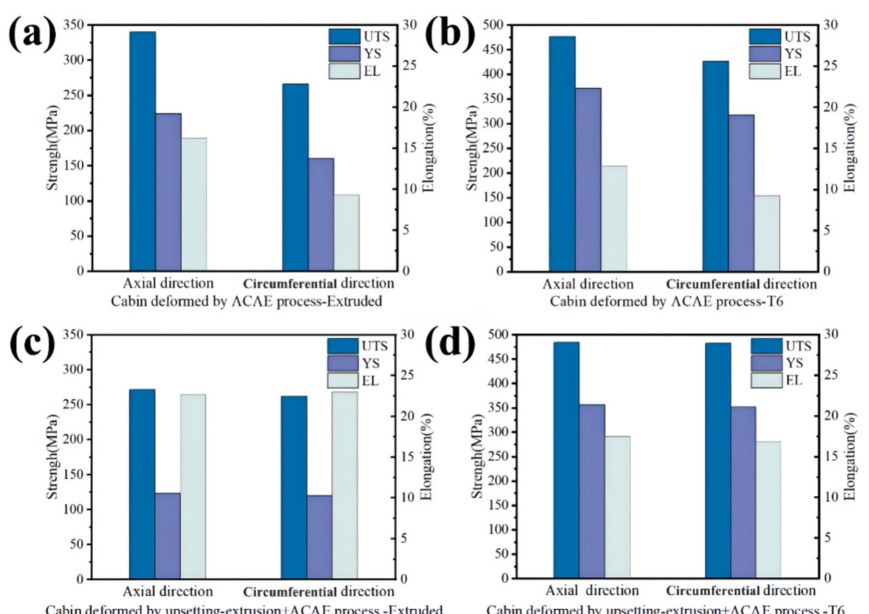

**Figure 10.** Tensile properties of Cabin Zone V in axial and circumscribed directions: (**a**) direct ACAE-extruded; (**b**) direct ACAE-T6; (**c**) UE + ACAE-extruded; (**d**) UE + ACAE-T6.

After, the billet was formed by the UE + ACAE process, and the extruded mechanical properties are shown in Figure 10c. The axial UTS is 271 MPa, the YS is 123 MPa, and the EL is 22.7%. The circumferential UTS is 262 MPa, the YS is 120 MPa, and the EL is 23%. The mechanical properties of the component after the T6 heat treatment are shown in Figure 10d. The axial UTS reaches 484 MPa, the YS is 356 MPa, and the EL is 17.5%. The circumferential UTS is 482 MPa, the YS is 351 MPa, and the EL is 16.8%.

The axial and circumferential mechanical properties of the cabin formed by the UE + ACAE process are more uniform. The lower UTS and YS of the extruded state are mainly due to the higher DRX ratio, which makes the material soften. The upsetting-extrusion process provides a more uniform grain size and fine dispersive second-phase particles for the component, which greatly enhances the deformation coordination ability of the material and may be beneficial to increase the degree of the solid solution and the nucleation sites that are precipitated by the aging in the subsequent solid solution and the artificial aging heat treatment. The results show that the mechanical properties of the components prepared by the UE + ACAE process are uniform in both the axial and circumferential directions in the extrusion state and the T6 state, and the EL increased at the small sacrifice of the YS.

## 4. Conclusions

The microstructure evolution and age-hardening behavior of 2A12 aluminum alloy components were studied by comparing predeformation (one-pass repetitive upsetting extrusion) and annular channel angular extrusion. The UE + ACAE process can significantly improve the mechanical uniformity and elongation of the 2A12 aluminum alloy ACAE component along the axial and circumferential mechanical properties. The results show that:

1.  Preforming the initial billet of the 2A12 aluminum alloy by the upsetting extrusion of several plastic deformations before the ACAE can effectively obtain the regular, fine, and dispersing second phases, and can increase the cumulative deformation, and obtain more refined grains;
2.  The second phases, with a regular shape and a fine and dispersive distribution, represent a higher ability to coordinate the deformation of the Al matrix, which plays a key role in improving the mechanical uniformity and elongation of the 2A12 aluminum alloy ACAE component along the axial and circumferential mechanical

properties. After the UE + ACAE process, the axial UTS and EL of the cabin in the extrusion state are 271 MPa and 22.7%, respectively. At the same time, the circumferential UTS and EL are 262 MPa and 23%, respectively. The UE process can significantly improve the mechanical properties of materials by affecting the second phase;

3.  After proper T6 heat treatment, the UE + ACAE-formed component can still maintain the characteristics of the uniform axial and circumferential mechanical properties and high elongation. After a solution treatment of 515 °C × 1 h, and an artificial aging treatment of 190 °C × 6 h, the 2A12 aluminum alloy cabin enjoys good mechanical properties and uniformity. The axial UTS is 484 MPa, and the EL is 17.5%. The circumferential UTS is 482 MPa, and the EL is 16.8%. The mechanical uniformity is better than in the ACAE process. High-performance cabins with a uniform performance can be prepared by combining the upsetting extrusion of several plastic deformations with the annular channel angular extrusion process.

**Author Contributions:** Conceptualization, methodology, X.Z. and L.-F.G.; software, writing—review and editing, K.C.; validation, resources, writing—original draft preparation, D.-K.W.; validation, Z.-M.Z. All authors have read and agreed to the published version of the manuscript.

**Funding:** This research was funded by the Key Core Technology and Generic Technology Research and Development Project of Shanxi province, Grant No. 2020XXX015.

**Institutional Review Board Statement:** Not applicable.

**Informed Consent Statement:** Not applicable.

**Data Availability Statement:** All data included in this study are available upon request by contact with the corresponding author.

**Conflicts of Interest:** The authors declare no conflict of interest.

## Appendix A

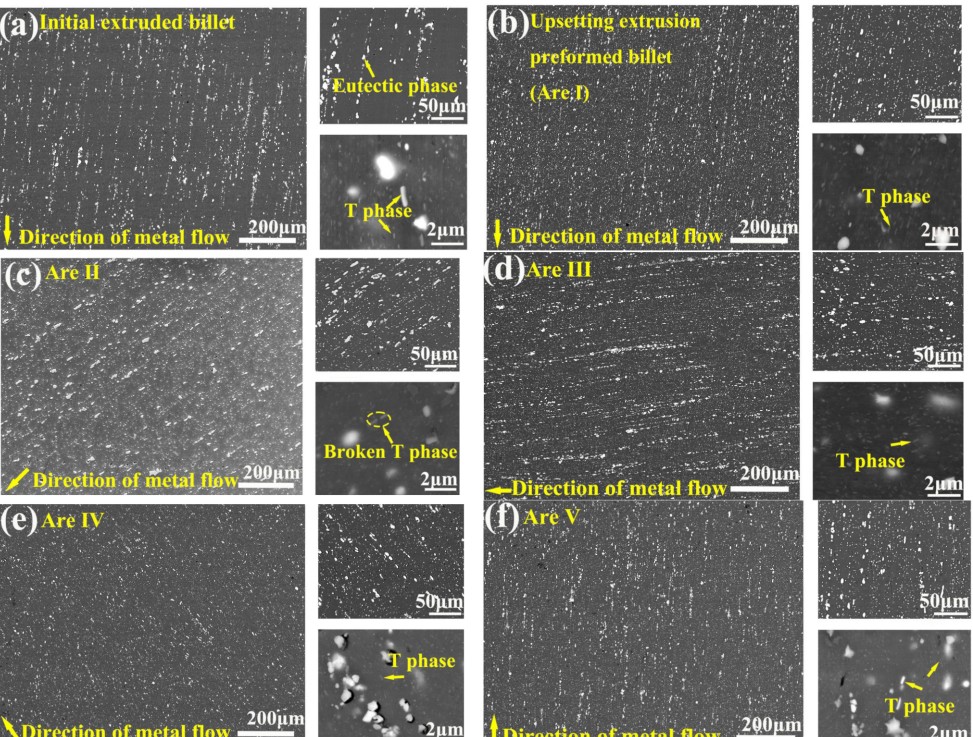

**Figure A1.** *Cont.*

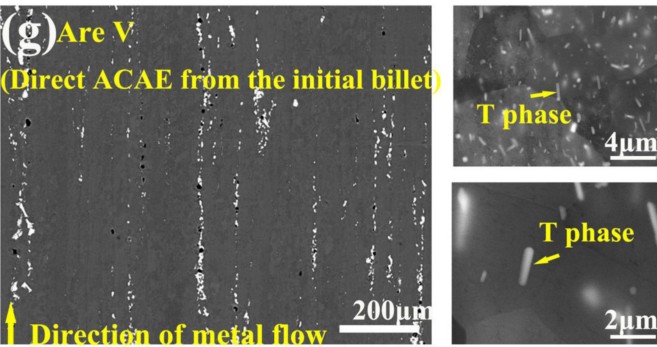

**Figure A1.** High resolution of Figure 4: (**a**) initial billet; (**b**) billet deformed by UE; (**c**) UE + ACAE Zone II; (**d**) UE + ACAE Zone III; (**e**) UE + ACAE Zone IV; (**f**) UE + ACAE Zone V; (**g**) ACAE Zone V.

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
