# Peer review of "Obtaining Uniform High-Strength and Ductility of 2A12 Aluminum Alloy Cabin Components via Predeformation and Annular Channel Angular Extrusion"

_coatings, doi:10.3390/coatings12040477_

Round 1
Reviewer 1 Report
Dear Authors
Your paper is generally well written, however before publication have to be corrected.
- Place in table the chemical composition, but the whole composition.
- Figure 1. Enlarge.
- Figure 2. Too small impossible to read measurements. Correct image quality.
- Figure 3. Correct scale quality. The mesh is invisible.
- Figure 4. In this figure it is impossible to see anything. In this form it can’t be presented.
- Figure 5. Too small figure, black letters on black background is not a good idea, impossible to read, very low quality of figure b).
- Figure 6. Low quality correct it.
- Figure 7. Too small.
- Figure 9 Very low quality and too small, black letters on black background is not a good idea.
- Lack of summary. I would add such part because it will help to understand all results and its comments.
- After adding summary conclusions also can be improved.
After corrections the paper can be published, but in this form it is impossible.
Reviewer 2 Report
The manuscript by Chen at evaluates the obtaining of uniform-high strength and ductility of 2A12 aluminium alloy cabin components via pre-deformation and annluar channel angular extrusion. In view of the data presented in this manuscript, it does not fit to the submitted special issue - Corrosion and degradation of materials. There is no link between the shown data to corrosion fundamentals, high-temperature oxidation, anodic oxidation, biochemical corrosion, stress corrosion cracking, corrosion fatigue, and corrosion creep, corrosion control and protection, or to coatings and surface characterization techniques. The only link could be the irregular/inhomogenous component phases (ln 156) as source of crack initiation and corrosion, or the heat treatment in section 3.3 (ln 254). However, only the first aspect is very briefly discussed. As such, in the present state the article would better fit another dedicated special issue or as a regular submission. Nevertheless, if the authors significantly improve the article to focus on aspects pertaining to the special issue, it would be than suitable.
In addition, the following points should be further improved:
- Figure 4 - BSE images. It is difficult to read the text on the images, and moreover, the Are I to Are V are not specified at all in the text.
- Figure 4b - the text on the EDX plot should be increased. Also figure caption should explain in detail what a, b represent.
- Figure 6 - scale bars are difficult to read. Similar for figure 8.
Round 2
Reviewer 1 Report
Dear Authors
Your paper is now better but I have two little suggestions
- Is in text Table 1. Chemical compositio. Should be of course Chemical composition. I would also add values of elements according to standard.
- Figure 4 add as a supplementary in high resolution at the end of file after References
Reviewer 2 Report
The authors addressed almost all the points in the reviewer comments. Nevertheless, I would suggest the authors to add a statement in the introduction as to the importance and correlation of their work with corrosion. Namely, with respect to the importance of the UE+ACAE process in improving the irregular second phase (which is usually the source of cracks and corrosion), towards a more regular, fine and dispersing second phase, and increase the cumulative deformation.
Other minor things:
- please remove lines 132-134 (probably leftovers from the template)
- citations of figures, use same template (mostly used Fig X). Correct on ln 141, 203, 221, 223, 259, 286, 288, 295, 297
- Rephrase/correct the use of "figure" on line 284.
- Overall I recommend the authors to go thorugh the article for minor English corrections.
